# Chemical Profiling and Nutritional Evaluation of Bee Pollen, Bee Bread, and Royal Jelly and Their Role in Functional Fermented Dairy Products

**DOI:** 10.3390/molecules28010227

**Published:** 2022-12-27

**Authors:** Amira M. G. Darwish, Aida A. Abd El-Wahed, Mohamed G. Shehata, Hesham R. El-Seedi, Saad H. D. Masry, Shaden A. M. Khalifa, Hatem M. Mahfouz, Sobhy A. El-Sohaimy

**Affiliations:** 1Food Industry Technology Program, Faculty of Industrial and Energy Technology, Borg Al Arab Technological University, Alexandria 21934, Egypt; 2Food Technology Department, Arid Lands Cultivation Research Institute, City of Scientific Research and Technological Applications (SRTA-City), Alexandria 21934, Egypt; 3Department of Bee Research, Plant Protection Research Institute, Agricultural Research Centre, Giza 12627, Egypt; 4Food Research Section, R&D Division, Abu Dhabi Agriculture and Food Safety Authority (ADAFSA), Abu Dhabi 52150, United Arab Emirates; 5International Joint Research Laboratory of Intelligent Agriculture and Agri-Products Processing, Jiangsu Education Department, Jiangsu University, Nanjing 210024, China; 6Pharmacognosy Group, Department of Pharmaceutical Biosciences, Uppsala University, Biomedical Centre, P.O. Box 591, SE 751 24 Uppsala, Sweden; 7International Research Center for Food Nutrition and Safety, Jiangsu University, Zhenjiang 212013, China; 8Department of Chemistry, Faculty of Science, Menoufia University, Shebin El-Koom 32512, Egypt; 9Department of Plant Protection and Molecular Diagnosis, Arid Lands Cultivation Research Institute, City of Scientific Research and Technological Applications (SRTACity), Alexandria 21934, Egypt; 10Abu Dhabi Agriculture and Food Safety Authority, Al Ain 52150, United Arab Emirates; 11Department of Molecular Biosciences, The Wenner-Gren Institute, Stockholm University, S-106 91 Stockholm, Sweden; 12Department of Plant Production, Faculty of Environmental Agricultural Sciences Arish University, North Sinai 45511, Egypt; 13Department of Technology and Organization of Public Catering, Institute of Sport, Tourism and Services, South Ural State University, 454080 Chelyabinsk, Russia

**Keywords:** honeybee products, amino acid profile, chemical profiling, international molecular network GNPS database, antioxidant potentials, proteolytic activity, fortified fermented milk

## Abstract

Honeybee products, as multicomponent substances, have been a focus of great interest. The present work aimed to perform the nutritional and chemical profiling and biochemical characterization of bee pollen (BP), bee bread (BB), and royal jelly (RJ) and study their applications in the fortification of functional fermented dairy products. Their effects on starter cultures and the physicochemical and sensorial quality of products were monitored. A molecular networking analysis identified a total of 46 compounds in the three bee products that could be potential medicines, including flavonoids, fatty acids, and peptides. BB showed the highest protein and sugar contents (22.57 and 26.78 g/100 g), which cover 45.14 and 53.56% of their daily values (DVs), with considerable amounts of the essential amino acids threonine and lysine (59.50 and 42.03%). BP, BB, and RJ can be considered sources of iron, as 100 g can cover 141, 198.5, and 94.94% of DV%, respectively. BP was revealed to have the highest phenolic and flavonoid contents (105.68 and 43.91 µg/g) and showed a synergetic effect when mixed with RJ, resulting in increased antioxidant activity, while BB showed a synergetic effect when mixed with RJ in terms of both antioxidant and proteolytic powers (IC_50_ 7.54, 11.55, 12.15, 12.50, and 12.65 cP compared to the control (10.55 cP)), reflecting their organoleptic properties and highlighting these health-oriented products as promising natural products for human health care.

## 1. Introduction

The development of advanced analytical tools and their applications have opened new possibilities to expand knowledge in the food science field. Bioactive compounds are a key challenge in the development of interventions involving tailored functional diets [1]. Mass Spectrometry combined with the international molecular networking GNPS database of food metabolite data can be a powerful resource to understand the molecular landscape and enables the reporting of research results in the context of foods [2,3,4].

Honeybee colonies produce various products, such as honey, bee pollen (BP), bee bread (BB), royal jelly (RJ), propolis, beeswax, and bee venom. BP, as a source of protein for the honeybee colony, is a highly nutritious substance that is commonly used to meet the growth and development needs of honeybees, including feeding broods and supplying the protein required for RJ secretion [5]. Honeybee workers collect BP from plant anthers and then place it in baskets (corbiculae) that are situated on their hind legs. In the composition of BP, there are more than 250 identified compounds, including amino acids, vitamins, macro- and micronutrients, and flavonoids [6,7]. BB is created from collected BP that is dampened with bee workers’ saliva, fragmented, packed in honeycomb cells, and then covered with a thin layer of honey, after which it is fermented and preserved by rising lactic acid under anaerobic conditions, which increases its nutritional value, digestibility, and absorption by the human body [8]. Additionally, BB contains considerably large amounts of peptides and all essential amino acids, which make it an excellent food product that could supplement deficient nutrients and also function to help eliminate various toxins [8]. RJ is secreted by nurse bee (6–12 days old) hypopharyngeal glands, playing an important role in larval development. Worker bee larvae are fed with RJ for 3 days, and only larvae that develop into queen bees and adult honeybee queens are continuously fed with large quantities of RJ [9]. RJ has been utilized in the pharmaceutical and food industries, as it provides essential amino acids, lipids, vitamins, acetylcholine (Ach), and other nutrients and active compounds of larval and adult queen food [10].

Functional foods are one of the growing food industries, where proteases are in great demand. Proteases are master enzymes in the digestive system, anatomically beginning at the oral cavity, proceeding to the esophagus, stomach, and intestinal tract, and ending with the colon [11]. The consequence of the proteolytic action of these enzymes is a change in the molecular conformation of native proteins to produce functional bioactive products that are widely used in food systems as additives [12]. These bioactive peptides can improve the functional and nutritional properties of proteins. They may aid in flavor and reduce the milk allergens in dairy products. Studies have indicated that some dietary proteins hydrolyzed by proteases show greater biological effects; for example, RJ peptides hydrolyzed by various proteases have been reported to show functional activities. The antioxidant activities derived from Major Royal Jelly Proteins (MRJPs) and peptides include scavenging activities, cholesterol-lowering effects, and anti-hypertensive ability and support the ongoing applications of available proteases [13,14]. However, it is difficult to know whether these enzymes are produced by the bee and secreted in RJ or whether they are of vegetal origin, which encourages more investigations in this area [15].

The objective of this study was the assessment of the nutritional profiles, chemical profiles, antioxidant potentials, and functional proteolytic enzymatic activities of honeybee products (BP, BB, and RJ), in addition to the evaluation of the effect of their combinations on their activities and their application in functional fermented dairy products by monitoring their impacts on starter cultures and consumer acceptance. This would pave the way to encouraging manufacturers to use fortifications with honeybee products in healthy functional food products as part of our ongoing project on honeybees and bee products [16,17,18].

## 2. Results and Discussion

### 2.1. Chemical Composition of Honeybee Products

The chemical compositions of BP, BB, RJ, and their mixture are exhibited in Table 1. BP showed a high content of total solids (TS) at 82.59% and was rich in protein and total sugars (21.09 and 19.69 g/100 g, respectively). Its ash (2.35%) was found to consist mainly of K, P, Ca, Na, and Mg in descending order (with values of 480.41, 226.15, 138.31, 98.65, and 96.41 mg/100 g, respectively), small amounts of Fe, Zn, and Mn, and traces of Cu. The results are in agreement with [7]. The main outcome measure of nutrient-based standards is whether the food can fulfill daily macro- and micronutrient requirements in a diet [19]. In this context, BP can play role in the fulfillment of daily protein and sugar requirements as macronutrients. On the other hand, it provides generous amounts of minerals, especially Fe and Mn [20].

Since BB is bee-collected pollen with a mixture of honey and bee salivary enzymes stored inside the beehive [21], its protein content was found to be comparable to that of BP (22.57 g/100 g), with higher contents of moisture and sugars (25.92 and 26.78 g/100 g, respectively). Compared to BP, BB contains more reducing sugars due to the effect of glycolytic enzymes, which convert complex sugars to simple sugars [22]. The potassium content in BB can provide 3.25% of DV with a value of 518.77 mg/100 g, followed by Ca, P, Na, Mg, and Fe. According to the required daily values of minerals, BB can be considered a rich source of Fe, Mn, and Cu since it provides 198.5, 91.5, and 70% of DV, respectively. Naturally, BB has various compositions depending on its BP source; however, the obtained nutrient composition is in agreement with that reported by [23].

Water was found to be the main component of RJ with a value of 63.81%, while its dry matter is composed mainly of carbohydrates (sugars) and proteins (14.78 and 12.89 g/100 g, respectively), which fulfill 29.56 and 25.78% of the daily requirements (Table 1). The predominant elements in descending order are K, P, Na, Ca, and Mg, with values of 124.01, 104.44, 74.42, 52.44, and 30.13 mg/100 g, respectively (Table 1). In addition, it can provide 17.09 and 0.50 mg/100 g of iron and copper, which cover 94.94 and 25% of the daily requirements, respectively. These results agree with [10,24], who also stated that the quantitative presence of RJ metals is related either to factors outside the colony, including the environment, food, and production period, or to internal factors, such as physiological factors tied to the nurse worker bees. From the nutritional perspective, which was supported by the obtained results, the combination of the three tested honeybee products (BP, BB, and RJ in a tri-mix) is considered useful by providing balanced contents of protein and sugars (37.38 and 39.38% of DV, respectively) and compensating for the lack of some minerals in the individual components, which encourages its application in functional foods’ fortification. The differences in the concentration levels of macro- and micronutrients in BP, BB, and RJ may be due to the nature of the sample and different environmental and physiological factors, such as the source of pollen, the nature of bee feeding, enzymes, and metabolic reactions in bee organs. In all cases, BP, BB, and RJ are considered good sources of proteins, sugars, and minerals.

### 2.2. Amino Acid Composition and Score (AAS%)

The amino acid profiles and amino acid scores (%) of BP, BB, and RJ are mentioned in Table 2. The term “complete protein” refers to foods that contain all nine essential amino acids in the proportions required to build proteins in the body. In contrast, “incomplete protein” refers to foods that have all essential amino acids but not in the correct proportions and is termed “limiting amino acid” [25]. The amino acid profiles of the three tested honeybee products (Table 2) reflect the high quality of protein, as they are shown to be significant sources of complete protein with a good balance of essential amino acids that achieve full protein adequacy in adults, especially in vegetarian/vegan diets. The main essential amino acids in BP were leucine, lysine, phenylalanine, + tyrosine, and threonine, with contents of 5.34, 5.04, 5.02, and 4.17 mg/g, covering 38.13, 42.03, 35.86, and 59.50%, respectively, for adult daily requirements, according to [26].

Additionally, BP contained considerable amounts of glutamic acid and proline (9.01 and 6.63 mg/g). Although neither glutamine nor proline is traditionally considered essential in the human diet, they are required in increased amounts in some pathological conditions and thus are usually classified as conditionally essential. Glutamine supplementation has been investigated in clinical work and sports nutrition; consequently, glutamine is readily available in many forms, ranging from pure glutamine powder to glutamine-fortified drinks and energy bars. Proline, however, has received very little attention; proline supplements (up to 488 mg/kg has been used to treat patients with gyrate atrophy due to a lack of ornithine aminotransferase) are not associated with reports of any deleterious side effects. However, it is generally accepted that the majority of proline synthesis in the body occurs via the glutamate/P5C synthase pathway [29]. BP is well-known as a rich source of proteins and essential amino acids with many pharmacological functions [30]. BB had high contents of phenylalanine + tyrosine, leucine, lysine, and methionine + cysteine (9.53, 9.46, 9.28, and 5.44 mg/g protein) that cover nearly 25% of AAS (26.58, 24.80, 22.07, and 26.58%, respectively). On the other hand, BB was revealed to contain notable amounts of proline and glutamic and aspartic acids (41.32, 39.32, and 14.74 mg/g, respectively). Egyptian BB had higher contents of amino acids than Malaysian BB according to data reported by [21]. RJ was found to contain good amounts of threonine, phenylalanine + tyrosine, lysine, and histidine, covering 23.54, 18.13, 18.05, and 17.07% of AAS, respectively. RJ contained limited amounts of non-essential amino acids, as aspartic acid had the highest content (11.56 mg/g). The obtained data on the amino acid composition of RJ are in agreement with [31]. Notably, both BP and RJ showed a deficiency of sulfur-containing amino acids, namely, methionine + cysteine, as they were shown to be their limiting amino acids (2.63 and 1.13 mg/g), while they are adequate in BB, covering 26.85% of AAS. Consequently, mixing protein sources with complementary essential amino acids within the same meal may achieve the limit of complete protein with high quality in human diets, ensuring long-term adequacy [32].

### 2.3. LC–MS–MS Analysis and International Molecular Networking GNPS Database of Bee Product Metabolites

Royal jelly, BB, and BB are examples of bee products that are rich in active compounds, such as proteins, amino acids, enzymes, fatty acids, esters, peptides, phenolic compounds, minerals, and vitamins [8,33,34]. Many factors, including climatic, botanical, geographic, and storage conditions, have an impact on the nutritional and chemical compositions of bee products [35,36].

The bee products*’* metabolomic mass profiles were determined using Global Natural Products Social Molecular Networking (GNPS) and MS-MS data in positive ionization mode, as shown in Figure 1 and Table 3. Each metabolite is represented by a node in the GNPS network, where metabolites of similar classes are grouped together to form a single cluster. A total of 222 nodes of bee products (BP, BB, and RJ) were identified. BP metabolites are demonstrated as pink-colored nodes, BB is blue, and orange nodes are RJ, as shown in Figure 1A.

Through LC-MS-MS analysis and molecular networking analysis, a total of 46 compounds were identified. These compounds included flavonoids, fatty acids, alkylglucosinolates, peptides, pyrrolizidine alkaloid, tannins, glycerophospholipids, steroids, epoxycarotenol, styrenes, phenols, glycerophospholipids, dihydrochalcone, and benzopyrans (Table 3). Sixteen parent ions matched known standards in the GNP library. The two main metabolites found in bee products, as listed in Table 3, are flavonoids and fatty acids. Flavonoids were found in all samples, and they were the most common class of natural products found (Figure 1B). Flavonoids were potent in BB and BB, whereas very few flavonoids were identified in the samples of RJ. Flavonoids appeared in the aglycone or flavonoid glycoside form as flavonol, flavone, and isoflavone. Since flavonoids possess anti-inflammatory, antibacterial, antifungal, antiviral, and anticancer activities, they are beneficial to human health [17,37,38]. RJ is distinguished from other bee products by the presence of an unsaturated fatty acid termed 10-hydroxy-2-decenoic acid (Table 3 and Figure 1C). This compound has anti-inflammatory and bactericidal effects on human colon cancer cells [39] and could be a potential medicine for rheumatoid arthritis [40]. The antibacterial compounds royalisin and jelleine II were found in RJ [41,42].

### 2.4. Antioxidant Potentials

Figure 2A–C present the antioxidant potentials of BP, BB, RJ, and their mixes, which are in accordance with the chemical profiling results. BP had the highest total phenolic content (TPC) (105.68 μg/g), followed by RJ and bee bread (66.35 and 56.93 μg/g) (Figure 2A). Just as the chemical profiling showed the potent contents of flavonoids in BP and BB, the total flavonoid content (TFC) assessment also revealed that BP had the highest contents of these compounds, followed by BB and then RJ (43.91, 20.49, and 15.29 μg/g, respectively) (Figure 2B).

The bi-mix (BP/RJ, BP/BB, and BB/royal jelly) and tri-mix results highlighted the role of BP in increasing TPC and TFC, which are correlated with the mixing ratios (1:1 or 1:1:1). On the other hand, while scavenging potentials, represented as IC_50_ (Figure 2C), of BP, bee bread, and RJ were recorded at 67.37, 39.84, and 63.10 mg/mL, the bi-mixes and tri-mix were found to have significantly higher antioxidant powers with lower IC_50_ values than those of the individual components. The IC_50_ values of the two BP bi-mixes were comparable (15.67 and 15.71 mg/mL), while the Tri-Mix showed less of an antioxidant effect with a higher IC_50_ of 31.43 mg/mL. Özkök & Silici (2017) reported similar results for BP, RJ, and their mixtures. The BB/RJ bi-mix showed remarkable results, as its IC_50_ had the lowest value (7.54 mg/mL); this significant result drew attention to another source of scavenging potentials besides phenolic and flavonoid contents, specifically enzymatic activity. Enzymatic hydrolysis was reported to improve the functional properties and biological activities of protein by-products, in addition to the DPPH radical-scavenging potentials of proteases that were previously reported by [12,69]. In order to support this hypothesis, qualitative, quantitative, and SDS-PAGE analyses were performed to assess the proteolytic activities of honeybee products and their mixes.

### 2.5. Protease Activity of Honeybee Products

Honeybee products’ crude extracts and their mixes were evaluated qualitatively and quantitatively for their potential proteolytic activities, and the results are illustrated in Table 4. In qualitative screening, royal jelly, BP/RJ, BB/RJ, and the tri-mix (BP/BB/RJ) exhibited positive proteolytic activity on casein agar (pH 7), with diameters of 1, 0.5, 0.5, and 0.5 mm, respectively (Table 4 and Figure 3). When comparing the screening results of the quantitative analysis at pH 7, RJ (5.22 U) showed approx. two times the proteolytic activity of BP and bee bread (2.20, 2.73 U), as the latter two bee products’ activities could not be revealed via screening by the qualitative method.

On the other hand, mixing RJ with BP significantly decreased its activity to 4.23 U. On the contrary, royal jelly mixed with bee bread significantly enhanced its activity to 6.52 U, which may be attributable to the composition of BB, which harbors bee salivary enzymes that showed a symbiotic effect with royal jelly. The tri-mix of the three tested bee products was equal to RJ alone (5.21 U) due to the two above-mentioned reverse interactions. The results reveal that pH is a crucial factor in protease activity. Due to the variations in optimal enzymatic protease activities, BB/RJ and the tri-mix showed the highest enzymatic activities at different pH values [70]. The highest proteolytic activity at pH 5.0 was exhibited by the BB/RJ extract (5.72 ± 0.07U). The tri-mix and BP/RJ showed activities of 4.35 ± 0.13 U and 3.23 ± 0.15 U, respectively. These values differed statistically based on a Duncan test at 0.05 (Table 5). Proteases with enzymatic activity optima at pH 5.0 could be used to coagulate milk proteins in the dairy industry, as debittering agents in cheese, and in peptide synthesis [71]. The evaluated samples that showed their highest proteolytic activities at pH 9 were BB/RJ (8.4 ± 0.17 U), RJ (7.28 ± 0.17 U), and the tri-mix (7.11 ± 0.16 U) (Table 5). The presence of proteases in RJ has been previously hypothesized and discussed, but only a trypsin-like protease and two serine-proteinases in RJ derived from thoracic glands were identified [15]. Matsuoka et al. (2012) indicated that RJ proteins consumed by honeybee queen larvae are hydrolyzed by enzymes in the larvae that are alkaline in nature, as these activities were stronger in the alkaline range at pH 9, at which most bands were hydrolyzed. This could explain the obtained results at pH 9.0 in the current research, which showed significant increases in proteolytic activities in bee products and their mixes, which indicates the alkaline nature of these proteases. These findings may provide us with novel knowledge that could be used in food technology as ingredients in functional foods and nutraceuticals [13].

### 2.6. SDS–PAGE of Honeybee Products

The electrophoresis patterns of honeybee products’ and their mixes’ extracted proteins in aqueous extracts are shown in Figure 4. SDS-PAGE showed bands with molecular weights ranging from ~10 to 118 kDa. Based on the band width and intensity, the major polypeptides present were in the range of 20–85 kDa. Proteins with MWs of 20, 26, 36, 47, and 77 KDa were found in all samples, while the protein with an MW of 23.6 KDa in BB and RJ corresponds to papain [72,73]. One protein band less than 20 KDa, corresponding to albumin components according to [74], was found in all samples with very high expression. It is noteworthy that the tri-mix (seventh lane) showed all band widths with high intensities, which agrees with the concept of mixing complementary protein sources to achieve a better balance of amino acids.

### 2.7. Physicochemical Characteristics of Developed Functional Fermented Dairy Products

Fermented dairy products fortified with PB, BB, RJ, and the tri-mix and the control are exhibited in Figure 5, and Table 5 illustrates the physicochemical characteristics of fortified fermented dairy products. The results revealed that compared to plain control yogurt (pH = 4.93), fortification with bee products and their mixture significantly decreased the pH to 4.83, 4.62, and 4.62 in TBB, TRJ, and TMix, respectively, and increased the titratable acidity from 0.350% in plain fermented milk to 0.399% in fortified fermented dairy. This decrease may be due to either bee products’ synergistic effect on the microbial fermentation of starter cultures or enzymatic activity producing amino and fatty acids that could affect acidity. These results disagree with the results of [75], who reported no significant changes in the acidity of fresh bio-yogurt fortified with RJ. The addition of BP, BB, RJ, and their mixture significantly improved the product texture compared to the control, as the viscosity significantly increased from 10.55 cP in control yogurt to 11.55, 12.15, 12.50, and 12.65 cP, respectively (*p* < 0.05). These results agree with [76], who reported that the fortification of yogurt with BP and RJ enhanced the textural properties. Additionally, Karabagias et al. reported that ground BP may act as a surface or interface enhancer in material or food science based on its total protein content. The obtained results emphasized that the fortification of fermented milk with honeybee products improved the physical properties of the products [77].

### 2.8. Effects of Fortification on LAB Starter Culture

The effects of fortification on starter cultures in fermented dairy products are illustrated in Figure 6. The main aim of microbiological analyses of fortified fermented dairy products was to ensure that these fortifications do not represent obstructions to the viability of lactic acid bacteria starter cultures, represented by *St. thermophiles* and *Lb. delbrueckii* subsp*. bulgaricus*, in addition to the assessment of the best-before date. Figure 6A describes the total counts in fermented dairy products over 15 days of cold storage. The counts were recorded at 7.7 × 10, 7.3 × 10, 1.42 × 10^2^, 1.63 × 10^2^, and 1.08 × 10^2^ CFU/g in the control, TBP, TBB, TRJ, and TMix, respectively, at zero time. The results agree with Nowak’s observations that the *Lactobacillus* genus showed higher counts in royal jelly samples and Hassan et al.’s finding that the viable counts of probiotics were boosted by fortification with RJ [78,79].

These results reflect the pH results (Table 5), which showed significantly lower pH in the fortified products TBB, TRJ, and TMix. The total counts then gradually increased until the 10th day of storage to 1.12 × 10^2^, 1.35 × 10^2^, 1.85 × 10^2^, 1.86 × 10^2^, and 1.96 × 10^2^ CFU/g, respectively, before starting to decrease on the 15th day. The same pattern was observed for the enumeration of *Streptococcus thermophilus* (Figure 6B) and counts of *Lactobacillus delbrueckii* subsp*. bulgaricus* (Figure 6C). The obtained results of the four forms of bee product fortification did not affect the total viable counts of the Streptococci or Lactobacilli group compared to control unfortified fermented milk during the storage period until the 10th day. Additionally, yeast and molds were not detected in any treatments up to the 15th storage day, which indicates good hygienic conditions of processing. The obtained results disagree with those of Atallah, who showed decreased counts during cold storage [76]. However, the obtained results indicate that these fermented dairy products are best consumed before 15 days of cold storage.

### 2.9. Sensory Evaluation

The sensory evaluations of fresh fortified fermented dairy products are illustrated in Figure 7. Sensory evaluations showed differences among treatments in the color, odor, taste, texture, appearance, and overall acceptance of fresh products (zero time). Fermented dairy fortified with RJ (TRJ) and the tri-mix (TMix) scored the highest total scores, which exceeded that of the plain control product, which may be related to pH and viscosity characteristics (Table 5) that served to increase consumer acceptability by enhancing the taste and texture. On the other hand, higher levels of probiotic starter culture counts in TRJ (Figure 6) may enhance the sensory properties due to the produced flavor compounds. On the contrary, fermented products fortified with BP (TBP) and BB (TBB) scored the lowest total scores. This could be related to the distribution of grains in the products that negatively affected taste, texture, and appearance scores (6.9, 7.0, and 7.8 for TBP and 7.7, 7.4, and 7.1 for TBB, respectively). The obtained findings are in agreement with [75,76], who reported the improved sensory quality of RJ-fortified bio-yogurt without adverse effects on lactic acid bacteria counts and organoleptic properties.

## 3. Materials and Methods

### 3.1. Sampling and Extracting

Samples of BP from different botanical origins, BB, and RJ were collected between May and July of 2019 from a honeybee apiary located in the Experimental Farm, City of Scientific Research and Technological Applications, Alexandria, Egypt. Beehives were equipped with bottom-fitted BP traps (Figure 8A) to collect BP (pollen grains) (Figure 8B), while BB (Figure 8D) was collected manually from honeycombs (Figure 8C). For royal jelly (RJ) production, Laidlaw and Page’s grafting method was applied. Two colonies were selected. One was a breeding colony headed by a young open-mated queen to produce a sufficient number of eggs that provided young larvae for grafting at 24–36 h old (Figure 8E) [80].

The second colony was the rearing colony (Figure 8F); this colony had a large population of nursing worker bees and was dequeened 48 h before grafting young larvae. With a grafting tool, young larvae were grafted and moved from breeding colony combs to queen cups in the rearing colony. After 48 h, all grafted cups were collected, the larvae were moved (Figure 8G), and then fresh RJ was harvested (Figure 8H). In the field, collected RJ was kept in an ice box, and it was directly moved to the laboratory and kept at −20 °C until being tested and used in experiments. Similarly, BP and BB samples were stored in a deep freezer (−20 °C), while mixes of honeybee products were freshly prepared before analyses.

For extraction, bee pollen and BB were crushed in a mortar, and then samples were extracted individually in Milli-Q (Ultra Clear Water Purification Systems, Series 2000, Siemens, Washington, DC, USA) with a ratio of 1:10 *w*/*v*, blended using a vortex mixer for 10 min, and then centrifuged at 4000 rpm/15 min, while RJ was dissolved with the same ratio. Bi-mixes were prepared with equal portions (1:1) *w*/*w*, as was the tri-mix (1:1:1) *w*/*w*/*w*.

### 3.2. Chemical Compositions of Honeybee Products

Moisture content was determined according to AOAC 925.10 [81], and ash content was determined as described by AOAC 923.03 [81]. Total protein content was determined by the binding of Coomassie Brilliant Blue G-250 to protein, using bovine serum albumin as a standard [82]. The phenol-sulfuric acid colorimetric method, using a T80 UV/VIS spectrophotometer (PG Instrument Ltd., Lutterworth, UK) at 490 nm, was employed to quantify the total soluble-sugar concentration in honeybee products, as described by [83].

### 3.3. Mineral Contents of Honeybee Products

Samples were prepared by dry ashing before dissolving in dilute aqua regia (10 mL concentrated HNO3: HCl, 1:3); the solution was diluted with Milli-Q water up to 50 mL in a volumetric flask, and then mineral concentrations were determined using Atomic Absorption Spectrometry (AAS) according to the method of [84], except for phosphorus, which was determined colorimetrically using the method of [85].

### 3.4. Daily Values (DVs%)

Daily values (%) for nutrition labeling were calculated based on a daily intake of 2000 calories, which has been established as the reference for adults and children 4 or more years of age according to [86].

### 3.5. Amino Acid Compositions of Honeybee Products

Amino acid analysis was carried out using an Automatic Amino Acid Analyzer (AAA 400 INGOS Ltd.) using the performic acid oxidation method according to [87,88]. The required pattern values were in accordance with previously established values (FAO/WHO/UNU, 1985; FAO/WHO, 1973; Recommended Dietary Allowances—Food and Nutrition/ Board Commission on Life/Sciences National Research Council, 1989), and the Amino Acid Score (AAS%) was calculated as “Percentage of adequacy” as follows:(1)$$\text{Amino}\;\text{Acid}\;\text{Score}\;(\%)=\frac{\text{mg}\;\text{of}\;\text{amino}\;\text{acid}\;\text{in}\;1\;\text{g}\;\text{test}\;\text{protein}}{\text{mg}\;\text{of}\;\text{amino}\;\text{acid}\;\text{in}\;\text{requirement}\;\text{pattern}}\times 100$$

### 3.6. Liquid Chromatography–Mass Spectrometry (LC-MS-MS) Analysis

Extracts of bee products were analyzed using LC–MS–MS. A Shimadzu LC-10 HPLC instrument with a Grace Vydac Everest Narrowbore C_18_ column (100 mm × 2.1 mm i.d., 5 µm, 300 Å). LC-MS connected to an LCQ electrospray ion trap MS (Thermo Finnigan, San Jose, CA, USA) was utilized with a mass range of 200–5000 m/z. A 2 µL sample was injected using an autosampler. The solvents used were 95% H_2_O in formic acid (0.1%) (A) and 95% ACN in formic acid (0.1%) (B). Gradient elution ranged from 5% to 95% (B), followed by column conditioning to 5% (B) at a 300 µL/min flow rate. The elution time was 60 min.

Foundation 3.1 Xcalibur 3.1.6610 was used to analyze the data. Additionally, MS Convert from the ProteoWizard suite (https://proteowizard.sourceforge.io/download.html; accessed on 8 November 2022) was used to convert the raw data files to mzXML format. The Global Natural Products Social Molecular Networking (GNPS) online workflow was used to generate the molecular network [3,4]. The network’s spectra were then validated against the spectral libraries and literature data of GNPS.

Cytoscape software was used to analyze and edit the molecular networks. The parent mass of each node served as a label. A pie slice proportional to the number of MS/MS spectra for each parent mass was established, with colors designating the sources of the samples.

### 3.7. Antioxidant Potentials of Honeybee Products

#### 3.7.1. Total Phenolic Content

The total phenolic content was assessed using the Folin–Ciocalteu reagent according to [89]. The absorbance was measured at 760 nm using a T80 UV/VIS spectrophotometer (PG Instrument Ltd., UK) and was expressed as gallic acid equivalents (GAE) in milligrams per gram sample. Measurements were performed in duplicate.

#### 3.7.2. Total Flavonoid Content

The total flavonoid content of aqueous extracts was assessed using Sakanaka et al.’s [90] colorimetric method [90], reading the absorbance at 510 nm using a T80 UV/VIS spectrophotometer (PG Instrument Ltd., UK). The results were expressed as μg/g of sample.

#### 3.7.3. DPPH Scavenging Activity

The DPPH (2,2-diphenyl-1-picrylhydrazyl) assay was used as described by [91], reading the absorbance at 517 nm using a T80 UV/VIS spectrophotometer (PG Instrument Ltd., UK).

### 3.8. Screening and Quantitative Determination of Proteolytic Activity

The qualitative proteolytic activity was detected by casein hydrolysis on agar plates containing YNB medium supplemented with 0.5% casein, 0.5% glucose, and 2% agar (*w*/*v*-1), pH 7.0 [92]. The plates were incubated at 28 °C for 3 days. Enzyme activity was indicated by the formation of a clear zone around the wells after precipitation with 1 M HCl solution. A commercial protease solution at 0.001% (*w*/*v*-) was used as the positive control.

A proteolytic qualitative activity assay, using casein as the substrate, was performed according to the method described by [93], with some modifications. Enzyme activity was determined by incubating 250 µL of the culture supernatant with 500 µL of 1% (*w*/*v*) casein sodium salt in 50 mM buffer (pH 5.0, 7.0, and 9.0) for 2h/30 °C. The reaction was stopped by the addition of 375 µL of 20% (*w*/*v*-1) trichloroacetic acid (TCA). The tubes were placed in an ice bath for 30 min and then centrifuged at 5000× *g* for 15 min at 4 °C. Proteolytic activity was determined from the absorbance reading of the supernatant at 280 nm versus an appropriate blank. One unit (U) of enzyme activity is defined as the amount of enzyme that, under the assay conditions described, gives rise to an increase of 0.1 unit of absorbance (280 nm) in 60 min at 30 °C (T80UV/Vis spectrometer PG Instruments LDT, United Kingdom).

The qualitative protease activity was assayed quantitatively using different pH values of the substrate solution (5.0, 7.0, and 9.0). The substrate was prepared in three different 50 mM buffers: sodium citrate (pH 5.0), sodium phosphate (pH 7.0), and Tris-HCl (pH 9.0) [94].

### 3.9. SDS-PAGE

The polypeptide patterns of honeybee products (bee pollen, BB, and RJ) were analyzed using sodium dodecyl sulfate–polyacrylamide gel electrophoresis (SDS-PAGE) according to [95]. Honeybee products were diluted to 1 mg/mL with 50 mM Tris buffer (pH 7.4) containing 1 mM EDTA (Merck, Darmstadt, Germany) and further mixed 1:1 (v:v) with Laemmli buffer containing 10% dithiothreitol (DTT) (Sigma-Aldrich, Steinheim, Germany). Subsequently, the mixture was boiled for 3 min and centrifuged for 3 min at 6000 g. Samples and a See-Blue Standard (10 xL) were loaded onto a 12% SDS-PAGE gel and run with running buffer at 200 V for approximately 50 min. Finally, the gels were stained with Coomassie Brilliant Blue G-250 overnight and washed with a distaining solution (15% ethanol and 5% acetic acid) until protein bands became clearly visible in the colorless gel matrix.

### 3.10. Application in Fermented Dairy Products

#### 3.10.1. Preparation of Honeybee-Product-Fortified Fermented Dairy Products

Pasteurized cow milk (3% fat) was standardized using skimmed milk powder (SMP) to raise the solid non-fat (SNF) from 8.5 to 13%. A sugar solution containing 65% sugar (*w*/*v*) was prepared and pasteurized at 65 °C/30 min [96]. Milk was warmed to 42 ± 2 °C and divided into five equal portions: C; control plain fermented milk; TBP; fermented milk fortified with bee pollen grains; TBB; fermented milk fortified with BB; TRJ; fermented milk fortified with RJ; and TMix; fermented milk fortified with tri-mix (BP, BB, RJ (1:1:1)). Honeybee products were added for a fortification percent of 1% (*w*/*v*), and the sugar solution was added at 7% (*v*/*v*). The fortified milk was then inoculated with the commercial starter culture YO-MIX^®^ (Danisco, France) containing *Streptococcus thermophilus* and *Lactobacillus delbrueckii* subsp. Bulgaricus, with an inoculation level of 10 Direct Culture Units (DCU) for 100 L milk (the lowest recommended inoculation dose), was poured into 100 mL bottles and incubated at 42 ± 2 °C until coagulation at pH ~4.6 (about 4 h) and then cooled and stored at 4 °C [97].

#### 3.10.2. Physicochemical Characterization of Fortified Fermented Dairy Products

pH was measured using a pH meter (Adwa AD1030, Hungary), and the titratable acidity of fortified fermented dairy products was expressed as an equivalent percentage of lactic acid according to [98].

The viscosity of fermented dairy products was determined using a J.P. Selecta Wide Range Rotary Viscometer (Model STS-2011, Valencia, Spain), with spindle number L3 at 100 rpm/20 °C. Fortified fermented dairy samples were left for 10 min at room temperature and stirred for 40 sec before analysis, and results were recorded in centipoises (cP) after 50 s of shearing [99].

#### 3.10.3. Microbiological Analysis

The conventional dilution pour-plate technique was used to enumerate microbes in the products. For the total viable microbial count, NA (Biolife, Milano, Italy) was used; members of *Lactobacilli* sp. were grown on MRS agar (Biolife, Milano, Italy), members of *Lactococci* sp. were grown on M17 agar (HiMedia, Mumbai, India), and the enumeration of yeast and mold was performed on potato dextrose agar (Biolife, Milano, Italy), as described by Standard Methods for the Examination of Dairy Products [100]. The results are represented as colony-forming units (CFU/g).

#### 3.10.4. Sensory Evaluation

Ten panelists (six men and four women, aged between 27 and 50 years) conducted sensory evaluations of fresh honeybee-fortified fermented dairy products (control plain fermented dairy, fermented dairy fortified with bee pollen grains, fermented dairy fortified with BB, fermented dairy fortified with RJ, and fermented dairy fortified with the tri-mix) at Food Technology Dept., Arid Lands Cultivation Research Institute, SRTA-City, Alexandria, Egypt, as described by [101,102] with some modifications. The criteria for selection depended on their experience and background related to fermented dairy products. The samples, which were stored at 4 °C, were allowed to rest at room temperature (20 °C ± 2) for 10 min before the evaluation. The samples were evaluated using a 9-point Hedonic scale according to ISO 22935-3 | IDF, 2009 [103]. This scale consisted of the test parameters of color, odor, taste, texture, appearance, and overall acceptability, accompanied by a scale of nine categories: 1 = dislike extremely; 2 = dislike very much; 3 = dislike moderately; 4 = dislike slightly, 5 = neither dislike nor like, 6 = like slightly; 7 = like moderately; 8 = like very much; 9 = like extremely. The data presented are averages of n = 10 ± SD.

### 3.11. Statistical Analysis

Data are expressed as means of duplicates ± standard deviation (SD). Data were analyzed using one-way analysis of variance (ANOVA) for multiple comparisons using the Duncan test in the IBM SPSS Statistics 23 software program, where a probability of *p* < 0.05 was considered statistically significant [104].

## 4. Conclusions

A molecular networking analysis identified a total of 46 compounds in three bee products that could be potential medicines, including royalisin and jelleine II, with anti-inflammatory activity, anti-colon-cancer effects, and antioxidant potential. Functional fermented dairy products can be successfully employed as vehicles to ease and broaden the delivery of honeybee products. Bee pollen, bee bread, and RJ had no adverse effects on starter cultures or organoleptic properties. Mixing protein sources with complementary essential amino acids and functional properties may help to achieve products with high nutritional quality for healthier diets. Consuming honeybee-product-fortified fermented dairy on a daily basis may add valuable nutrients due to their high nutritional value, high protein quality, in vitro antioxidant capacities, and proteolytic activities in parallel with preferable sensorial aspects. In conclusion, the complexity of honeybee products in terms of their nutrient compositions and chemical, functional, and proteolytic properties can open future research opportunities of great importance for the development of functional dairy products on the industrial scale.

## Figures and Tables

**Figure 1 molecules-28-00227-f001:**
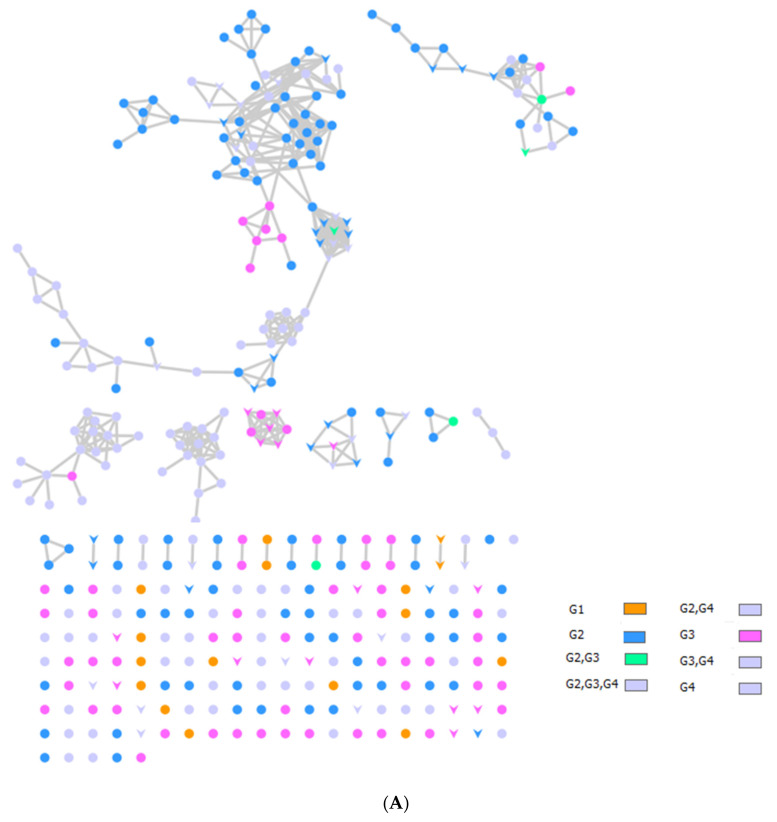
(**A**) Metabolite molecular network of the bee product (royal jelly, bee bread, and bee pollen) extracts and the blank injected before the extracts. The nodes refer to parent masses of the extract metabolites. The circular nodes refer to whole-parent masses that have unique detected peaks in the molecular network. The triangle nodes represent parent ions that have been identified in the GNPS molecular network. G1: royal jelly nodes are orange; G2: bee pollen is colored pink; G3: bee bread is blue; and G4: blank solvent. (**B**) Flavonoids found in bee product extracts, including royal jelly, bee bread, and bee pollen. (**C**) Fatty acid, glycerophospholipid, styrene, and phenol metabolites from bee product (royal jelly, bee bread, and bee pollen) extracts.

**Figure 2 molecules-28-00227-f002:**
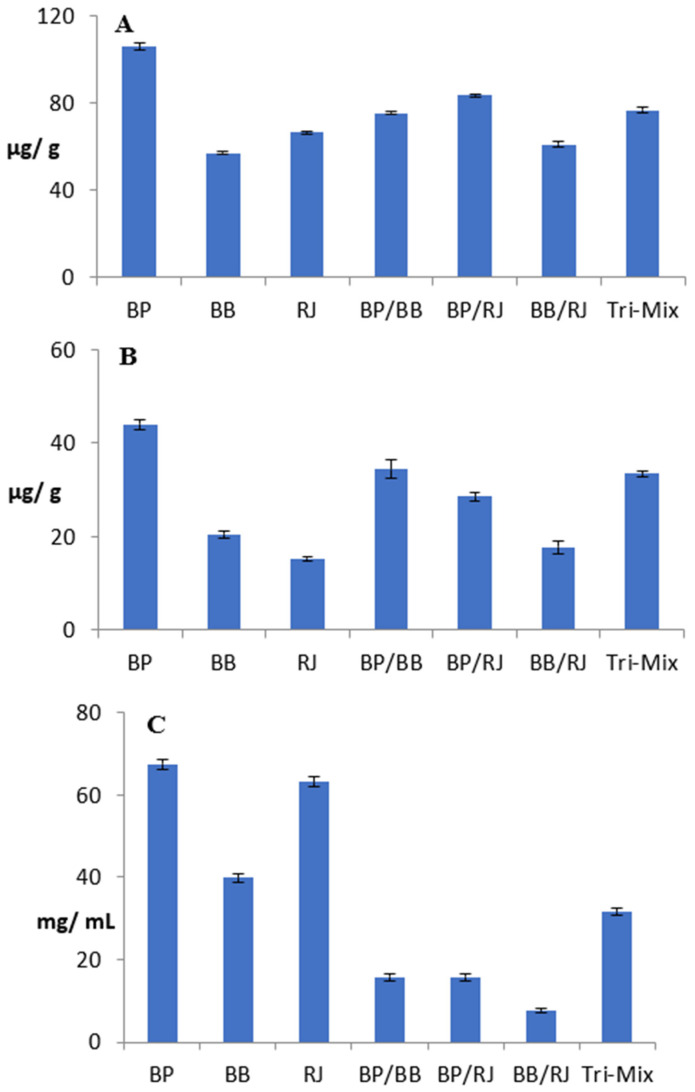
Antioxidant potentials of bee pollen, bee bread, royal jelly, and their mixes. (**A**): Total phenolic content (μg/g); (**B**): total flavonoid content (μg/g); (**C**): antioxidant potentials represented as IC50 (mg/mL), the inhibitory concentration at which 50% of DPPH radicals are scavenged. BP: bee pollen; BB: bee bread; RJ: royal jelly. Data represented are means of duplicates ± SD.

**Figure 3 molecules-28-00227-f003:**
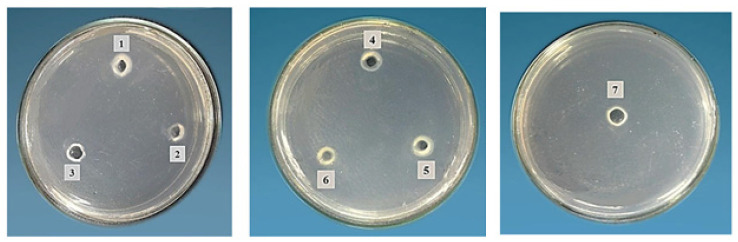
Proteolytic activity on casein agar of bee products: 1, bee pollen; 2, bee bread; 3, royal jelly; 4, bee pollen/bee bread; 5, bee pollen/royal jelly; 6, bee bread/royal jelly; 7, tri-mix.

**Figure 4 molecules-28-00227-f004:**
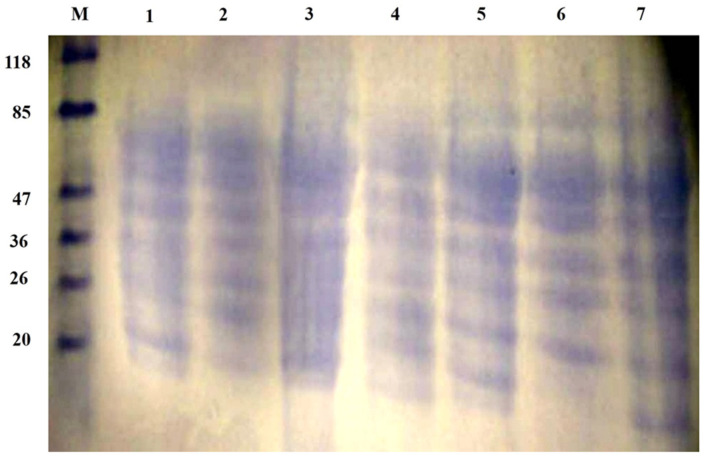
SDS-PAGE of honeybee products and their mixes. M, protein marker; 1, bee pollen; 2, bee bread; 3, royal jelly; 4, bee pollen/bee bread; 5, bee pollen/royal jelly; 6, bee bread/royal jelly; 7, tri-mix.

**Figure 5 molecules-28-00227-f005:**
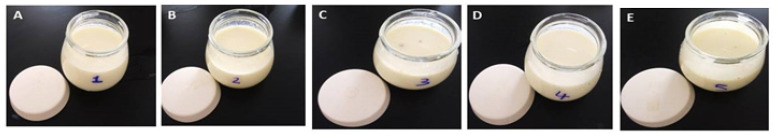
Fortified fermented milk products with different treatments. (**A**): Plain fermented milk (control); (**B**): fermented milk fortified with bee pollen (TBP), (**C**): fermented milk fortified with bee bread (TBB); (**D**): fermented milk fortified with RJ (TRJ); (**E**) fermented milk fortified with tri-mix (TMix).

**Figure 6 molecules-28-00227-f006:**
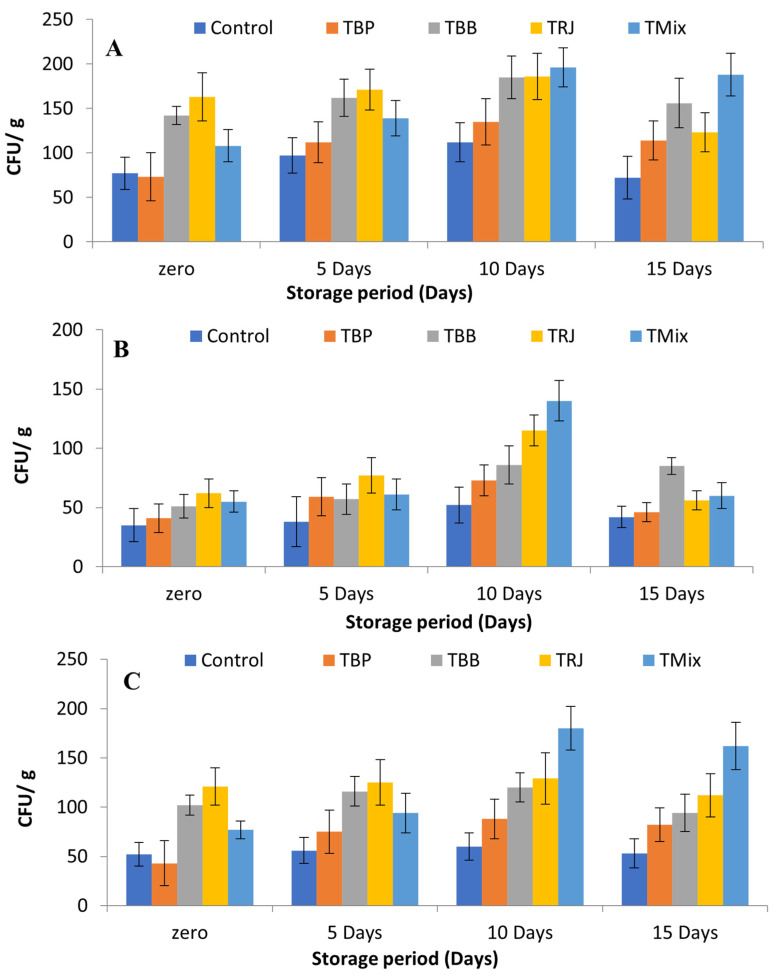
Effects of fortification on starter cultures in fermented milk products during cold storage. (**A**): Total counts on NA; (**B**): lactic acid bacteria counts on M17 for *Streptococcus thermophilus*; (**C**): lactic acid bacteria counts on MRS for *Lactobacillus delbrueckii* subsp*. Bulgaricus*. Data presented as means of duplicates ± SD. Control, plain fermented milk; TP, fermented milk fortified with bee pollen; TBB, fermented milk fortified with bee bread; TRJ, fermented milk fortified with royal jelly; TMix, fermented milk fortified with tri-mix.

**Figure 7 molecules-28-00227-f007:**
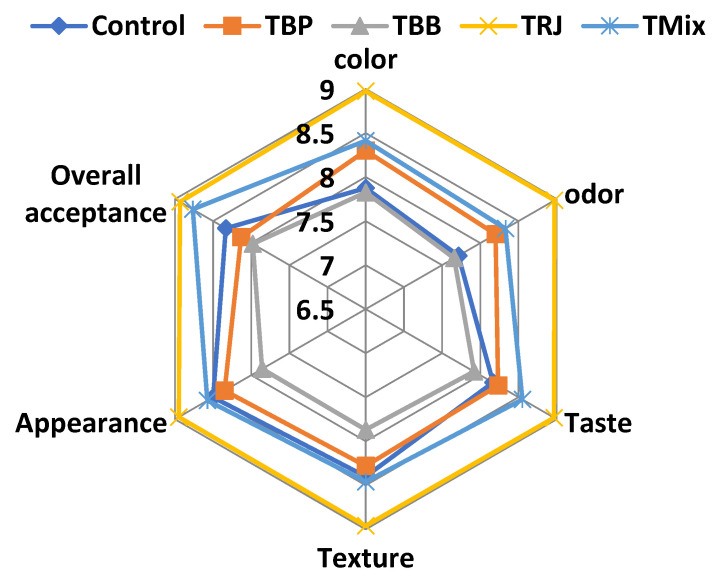
Sensory evaluation of fresh fortified fermented milk products. Control, plain fermented milk; TBP, fermented milk fortified with bee pollen; TBB, fermented milk fortified with bee bread; TRJ, fermented milk fortified with royal jelly; TMix, fermented milk fortified with tri-mi.

**Figure 8 molecules-28-00227-f008:**
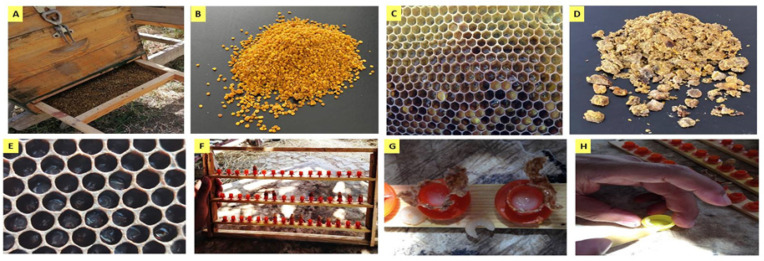
Bee products’ raw materials and grafting method for royal jelly production. (**A**) Bee pollen trap; (**B**) collected bee pollen; (**C**) bees’ honeycomb showing bee bread stored inside hexagonal bee cells; (**D**) collected bee bread; (**E**) young larvae less than 36 h old; (**F**) rearing queen cells; (**G**) moving larvae from grafted cups; (**H**) royal jelly harvesting.

**Table 1 molecules-28-00227-t001:** Nutritional profiles and daily values (%) of bee products and their mixture.

Component	Bee Pollen		Bee Bread		Royal Jelly		Tri-Mix	
	Content	^1^ DV%	Content	^1^ DV%	Content	^1^ DV%	Content	^1^ DV%
**Nutrients (g/100 g)**								
Moisture	17.41 ± 1.74	-	25.92 ± 0.14	-	63.81 ± 1.24	-	26.38 ± 0.58	-
Ash	2.35 ± 0.16	-	3.02 ± 0.07	-	3.25 ± 0.09	-	3.35 ± 0.19	-
Protein	21.09 ± 0.01	42.18	22.57 ± 0.07	45.14	12.89 ± 0.21	25.78	18.69 ± 0.05	37.38
Total sugars	19.69 ± 0.51	39.38	26.78 ± 1.80	53.56	14.78 ± 1.29	29.56	19.69 ± 2.06	39.38
**Minerals (mg/100 g)**								
Calcium, Ca	138.31	13.83	282.87	28.29	52.44	5.24	272.29	27.23
Copper, Cu	0.97	48.50	1.40	70.00	0.50	25.00	1.31	65.50
Iron, Fe	25.40	141.11	35.73	198.50	17.09	94.94	40.94	227.44
Potassium, K	480.41	13.73	518.77	14.82	124.01	3.54	504.71	14.42
Magnesium, Mg	96.41	24.10	85.13	21.28	30.13	7.53	80.20	20.05
Manganese, Mn	2.12	106.00	1.83	91.50	0.16	8.00	1.66	83.00
Sodium, Na	98.65	4.11	120.16	5.01	74.42	3.10	99.11	4.13
Phosphorus, P	226.15	22.62	192.25	19.23	104.44	10.44	241.00	24.10
Zinc, Zn	5.91	39.40	4.72	31.47	1.42	9.47	4.57	30.47

Data presented are the means of duplicates ± standard deviations. ^1^ DV% is the daily value for nutrition labeling, calculated per 100 g and based on a caloric intake of 2000 calories for adults and children 4 years or more.

**Table 2 molecules-28-00227-t002:** Amino acid profiles of bee pollen, bee bread, and royal jelly.

Amino Acid	Symbol	^1^ Pattern	Bee Pollen		Bee Bread		Royal Jelly	
			mg/g	^2^ AAS	mg/g	^2^ AAS	mg/g	^2^ AAS
**Essential Amino Acids**								
Histidine	His	12.00	3.35	27.91	4.66 *	16.71	2.85	17.07
Leucine	Leu	14.00	5.34	38.13	9.46	24.80	4.04	16.30
Isoleucine	Ile	10.00	2.93	29.32	4.97	16.95	2.21	13.04
Lysine	Lys	12.00	5.04	42.03	9.28	22.07	3.98	18.05
Methionine + cysteine	Met + Cys	13.00	2.63 *	20.26	5.44	26.85	1.13 *	4.22
Phenylalanine + tyrosine	Phe + Tyr	14.00	5.02	35.86	9.53	26.58	4.87	18.31
Threonine	Thr	7.00	4.17	59.50	6.33	10.64	2.50	23.54
Valine	Val	10.00	3.97	39.74	6.48	16.30	2.96	18.13
**Non-Essential Amino Acids**								
Alanine	Ala	-	5.94	-	7.79	-	1.99	-
Arginine	Arg	-	4.23	-	7.63	-	2.96	-
Aspartic acid	Asp	-	8.66	-	14.74	-	11.56	-
Glutamic acid	Glu	-	9.01	-	39.32	-	5.99	-
Glycine	Gly	-	4.39	-	6.71	-	1.80	-
Proline	Pro	-	6.63	-	41.32	-	4.22	-
Serine	Ser	-	4.76	-	9.37	-	3.00	-

^1^ Pattern (mg/g protein) for adults according to [26], based on highest estimate of requirement to achieve nitrogen balance (estimated amino acid requirements in adults) [27], assuming a safe level of protein intake of 0.55 g per kg per day (average value for men and women) [28]. ^2^ AAS: amino acid score (%). * First limiting amino acids.

**Table 3 molecules-28-00227-t003:** LC-MS/MS data of the annotated metabolites in bee product extracts.

No.	Compound Name	RT (min)	Parent Mass (g/moleg)	Molecular Formula	Fragments from the Raw Mass Spectrum (MS–MS of [M+]	Class	Reference
1	Apigenin-7-*O*-glucoside ^1^	0.41	433.110	C_21_H_20_O_10_	385.9840, 265.0950, 221.0030	Flavonoid	[43]
2	Glycerophosphorylcholine ^2^	1.50	258.240	C_8_H_20_NO_6_P	184.0270, 103.9920	Glycerophospholipids	https://bit.ly/3cCJup1 (accessed on 8 November 2022)
3	Linolenic acid ^3^	2.29	279.095	C_18_H_30_O_2_	205.0880, 149.0250, 135.1179	Fatty acid	[44]
4	(9*Z*,12*Z*,15*Z*)-Octadecatrienoic acid ^3^	2.29	279.679	C_18_H_30_O_2_	223.1450, 206.0780, 169.0890, 128.1160	Fatty acid	[45]
5	Gluconapin ^1^	6.69	373.312	C_11_H_19_NO_9_S_2_	133.1230, 113.0610, 85.0760	Alkylglucosinolate	[46]
6	Erucifoline ^2^	7.95	350.205	C_18_H_23_NO_6_	332.0370, 303.9810, 259.8500, 186.0020, 122.0490	Pyrrolizidine alkaloid	[47]
7	Galloyl-glucose ^2^	8.85	332.250	C_13_H_15_O_10_	314.0940, 286.0230, 242.0300, 167.9630	Tannins	[48]
8	Glucotropaeolin ^2^	9.48	410.478	C_14_H_19_NO_9_S_2_	284.0570, 206.0510, 163.9885, 149.0770,	Alkylglucosinolate	[46]
9	1-Palmitoyl-2-hydroxy-sn-glycero-3-phosphoethanolamine ^2^	10.10	469.383	C_21_H_44_NO_8_P	313.2810, 258.1590, 184.0740, 125.090, 104.1170, 86.1320, 71.1090,	Fatty acid	https://bit.ly/3Kx8UkB (accessed on 8 November 2022)
10	*Trans*-cinnamic acid ^3^	10.38	148.050	C_9_H_8_O_2_	149.0472, 131.9598	Cinnamic acid	[49]
11	Quercetin 3,4′-diglucoside ^2^	10.97	627.2240	C_27_H_30_O_17_	464.9100, 302.9680, 257.0180, 229.0340	Flavonoid	https://bit.ly/3CLx0G8 (accessed on 8 November 2022)
12	Myricetin-3-*O*-rutinoside ^2^	11.96	627.150	C_27_H_30_O_17_	478.9240, 463.0380, 316.9320,	Flavonoid	[50]
13	Kaempferol-7-*O*-neohesperidoside ^2^	11.97	595.204	C_27_H_30_O_15_	448.9148, 329.0170, 286.9450	Flavonoid	https://bit.ly/3KH5MTi (accessed on 8 November 2022)
14	1-Palmitoylglycerophosphocholine ^2^	15.14	497.621	C_24_H_51_NO_7_ P+	479.2050, 183.9420	Glycerophospholipid	https://bit.ly/3Tycvmz (accessed on 8 November 2022)
15	Kaempferol ^2^	15.63	287.234	C_15_H_10_O_6_	270.930, 240.9860, 231.0140, 212.9970, 164.9480, 152.930, 120.9760	Flavonoid	[51] https://bit.ly/3YMu2tH (accessed on 8 November 2022)
16	Luteolin ^2^	15.63	287.234	C_15_H_10_O_6_	240.9860, 176.9550, 152.9390	Flavonoid	[52,53]
17	Cholesterol ^3^	16.14	387.204	C_27_H_46_O	331.1450, 275.0680, 175,0180, 77.0670	Steroid	[54] https://bit.ly/3ABPp5R (accessed on 8 November 2022)
18	Erucic acid ^3^	16.48	339.220	C_22_H_42_O_2_	321.2420, 167.1070, 113.0830	Fatty acid	[55]
19	8 Methoxykaempferol ^2^	17.31	317.218	C_16_H_12_O_7_	299.0680, 182.9630,	Flavonoid	[56]https://bit.ly/3FP7CAz
20	Isorhamnetin-3,7-di-O-glucoside ^3^	18.55	641.199	C_28_H_32_O17	479.1630, 317.076, 145.0590, 127.0400	Flavonoid	https://bit.ly/3cxxxRp (accessed on 8 November 2022)
21	3-[(2*S*,3*R*,4*S*,5*S*,6*R*)-4,5-Dihydroxy-6-(hydroxymethyl)-3-[(2S,3R,4S,5S,6R)-3,4,5-trihydroxy-6-(hydroxymethyl)oxan-2-yl]oxyoxan-2-yl]oxy-2-(3,4-dihydroxyphenyl)-5-hydroxy-7-methoxychromen-4-one ^3^	18.66	641.199	C_28_H_32_O_17_	317.0720, 228.0740, 127.0500, 109.0410, 85.0560	Flavonoid	https://bit.ly/3cxQvaL (accessed on 8 November 2022)
22	**Dihydroquercetin** ^3^	19.40	305.208	C_15_H_12_O_7_	153.0180, 135.0550, 95.1010	Flavonoid	[57]
23	**Antheraxanthin** ^3^	19.99	584.340	C_40_H_56_O_3_	584.3281, 566.2841, 548.3191	Epoxycarotenol	[58]
24	Catechin ^3^	20.09	291.194	C_15_H_14_O_6_	273.1880, 109.0810	Flavonoid	[59,60]
25	Eicosanoic acid ^2^	20.28	313.391	C_20_H_40_O_2_	295.097, 226.9930, 113.0180	Fatty acid	[45]
26	Narcissin ^2^	20.40	625.140	C_28_H_32_O_16_	478.9240, 463.0380, 316.9320	Flavonoid	https://bit.ly/3R39BV2 (accessed on 8 November 2022)
27	Petunidin-3-*O*-rutinoside ^3^	20.40	627.170	C_28_H_35_O_16_+	627.3573, 480.1067, 317.0757	Flavonoid	[61]
28	1-(9Z,12Z-Octadecadienoyl)-sn-glycero-3-phosphocholine ^2^	24.56	520.603	C_26_H_50_NO_7_P	520.2020, 335.1590, 183.9490165.9880	Glycerophospholipid	https://bit.ly/3B1qeLd (accessed on 8 November 2022)
29	Pinobanksin5-methylether-3-*O*-acetate ^3^	20.80	314.202	C_18_H_17_O_5_	254.1670, 236.1810, 180.1590, 162.9690	Flavonoid	[59]
30	10-Hydroxy-2-decenoic acid ^1^	21.53	187.130	C_10_H1_8_O_3_	187.1488, 169.1582, 153.1280, 142.1416, 129.1183, 87.9283	Fatty acid	[62]
31	Jelleine II ^1^	21.79	1054.300	C_51_H_82_N_12_O_12_	1053.8247, 527.9614	Peptide	[42]
32	1-Decanoyllysolecithin ^3^	23.20	412.217	C_18_H_38_NO_7_P	229.1340, 184.0640, 166.0410, 125.000, 104.0930, 86.1210, 59.0810	Glycerophospholipid	https://bit.ly/3ednJwB (accessed on 8 November 2022)
33	*p*-Coumaroyl spermidine ^3^	23.26	291.184	C16H_25_N_3_O_2_	233.1550, 219.1350, 177.1480	Styrene	[63]
34	*p*-**Coumaroyl** glycerl ^3^	23.63	239.159	C_12_H_14_O_5_	221.1500, 206.5031, 198.1296	Phenol	[64]
35	Unknown ^1^	23.84	2586.186	-	1293.093, 863.3149, 647.7252, 518.5782, 432.4673	Polypeptide	-
36	Formononetin^3^	24.77	267.120	C_16_H_12_O_4_	131.0950, 119.0610	Flavonoid	[65]
37	Delphinidin-3-*O*-rutinoside ^3^	24.91	647.130	C_27_H_31_ClO_16_	484.2287, 322.1964, 308.2132, 291.1788, 177.0493	Flavonoid	[61]
38	[(2R)-2-Hydroxy-3-[(Z)-icos-9-enoyl]oxypropyl] 2-(trimethylazaniumyl)ethyl phosphate ^3^	25.97	549.379	C_28_H_56_NO_7_P	349.3070, 184.0731, 142.0910, 125.0070, 86.1240, 60.1180	Glycerophospholipid	https://bit.ly/3R8MTuW (accessed on 8 November 2022)
39	Royalisin ^1^	26.07	5532.370	C_240_H_378_N_66_O_72_S_6_	1382.8334, 1106.0650, 921.9670, 790.4330, 691.6823	Polypeptide	[41]
40	*N′*,*N″*,*N′″-Tris*-*p*-feruloylspermidine ^3^	27.420	674.130	C_37_H_43_N_3_O_9_	674.3249, 498.2758, 0.2445, 305.2053, 289.1925, 177.0558	Phenylamide	[66]
41	1-(9*Z*-Octadecenoyl)-sn-glycero-3-phosphocholine ^3^	28.85	520.363	C_26_H_52_NO_7_P	184.0800, 125.0050, 104.1150, 86.1280, 60.1070	Glycerophospholipid	https://bit.ly/3pWfJTh (accessed on 8 November 2022)
42	Phloretin ^2^	30.86	274.262	C_15_H_14_O_5_	256.000, 238.0010, 154.0300	Dihydrochalcone	[67]
43	Capric acid ^3^	39.20	171.098	C_10_H_20_O	153.0930, 125.1070, 97.1230, 83.1100, 67.0730	Fatty acid	[68]
44	[(2*R*)-3-Hexadecanoyloxy-2-hydroxypropyl] 2-(trimethylazaniumyl)ethyl phosphate ^3^	45.32	496.383	C_24_H_50_NO_7_P	478.2009, 183.9610	Glycerophosphocholine	https://bit.ly/3dZzCWC (accessed on 8 November 2022)
45	[(2*R*)-3-Octadecanoyloxy-2-[(Z)-octadec-11-enoyl]oxypropyl] 2-(trimethylazaniumyl)ethyl phosphate ^1^	47.22	787.830	C_44_H_86_NO_8_P	524.4550, 184.0970, 99.5210	Fatty acid	https://bit.ly/3RjZLxM (accessed on 8 November 2022)
46	6,6,9-Trimethyl-3-pentyl-6a,7,8,10a-tetrahydrobenzo[c]chromen-1-ol ^3^	52.55	313.236	C_21_H_30_O_2_	105.0820, 105.0820, 91.0670	Benzopyran	https://bit.ly/3RnuWIK (accessed on 8 November 2022)

^1^ Royal jelly; ^2^ bee bread; ^3^ bee pollen extracts, (-): not reported.

**Table 4 molecules-28-00227-t004:** Qualitative and quantitative assessment of protease activities for bee products and their mixes.

Sample	Casein Hydrolysis(Diameter mm)	Proteolytic Activity (U)		
		pH 5.0	pH 7.0	pH 9.0
Bee pollen	-	1.03 ± 0.14 ^fC^	2.20 ± 0.1 ^fB^	3.65 ± 0.098 ^eA^
Bee bread	-	1.22 ± 0.092 ^fC^	2.73 ± 0.12 ^eB^	4.73 ± 0.7 ^cA^
Royal jelly	1.0 ± 0.1 ^a^	2.27 ± 0.13 ^dC^	5.22 ± 0.11 ^bB^	7.28 ± 0.17 ^bA^
Bee pollen/bee bread	-	2.00 ± 0.1 ^eC^	3.22 ± 0.01 ^dB^	4.96 ± 0.15 ^cA^
Bee pollen/royal jelly	0.5 ± 0.1 ^b^	3.23 ± 0.15 ^cC^	4.23 ± 0.12 ^cB^	4.29 ± 0.13 ^dA^
Bee bread/royal jelly	0.5 ± 0.1 ^b^	5.72 ± 0.07 ^aC^	6.52 ± 0.11 ^aB^	8.4 ± 0.17 ^aA^
Tri-mix	0.5 ± 0.2 ^b^	4.35 ± 0.13 ^bC^	5.21 ± 0.1 ^bB^	7.11 ± 0.16 ^bA^

Data represented as means of duplicates ± SD. Means followed by the same letters did not differ from one another according to the Duncan test (*p* < 0.05). Capital letters compare enzyme activities at different pH values in the same rows, and lowercase letters compare enzyme activities for each sample in the same column. Proteolytic unit (U): one unit (U) of enzyme activity is defined as the amount of enzyme that produces an increase of 0.1 unit of absorbance (280 nm) in 60 min/30 °C.

**Table 5 molecules-28-00227-t005:** Physicochemical characteristics of fortified fermented milk products.

Sample	pH	TA%	Viscosity (cP)
Control	4.93 ± 0.02 ^a^	0.350 ± 0.01 ^b^	10.55 ± 0.07 ^e^
TBP	4.90 ± 0.01 ^a^	0.399 ± 0.01 ^a^	11.55 ± 0.21 ^d^
TBB	4.83 ± 0.01 ^b^	0.399 ± 0.01 ^a^	12.15 ± 0/07 ^bc^
TRJ	4.62 ± 0.01 ^c^	0.399 ± 0.01 ^a^	12.50 ± 0.14 ^b^
TMix	4.62 ± 0.03 ^c^	0.399 ± 0.01 ^a^	12.65 ± 0.21 ^a^

Data presented are the means of duplicates ±SD. TA, titratable acidity expressed as % lactic acid; cP, centipoise. Means followed by different letters in each column are significantly different at *p* < 0.05. Control, plain fermented milk; TBP, fermented milk fortified with bee pollen; TBB, fermented milk fortified with bee bread; TRJ, fermented milk fortified with royal jelly; TMix, fermented milk fortified with tri-mix.

## Data Availability

Not applicable.

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
