# Peer review of "Chemical Profiling and Nutritional Evaluation of Bee Pollen, Bee Bread, and Royal Jelly and Their Role in Functional Fermented Dairy Products"

_molecules, 2022, doi:10.3390/molecules28010227_

Round 1

Reviewer 1 Report

The manuscript molecules-2053574, entitled “Chemical profiling and Nutritional evaluation of bee pollen, bee bread, royal jelly and their role in functional fermented dairy products” is a good research about the chemical and biological properties of these bee products and their incorporation to fermented dairy products. However, the manuscript needs to be improved because I have lost in some part and it has some errors.

Comments to improve the manuscript:

In general, along the text there are different sentences between parentheses that should not be in parentheses, i.e. “such as (the source of the pollen, the nature of bee feeding, the enzymes and metabolic reaction…”, also some ºC are wrong written.

Abstract:

There are some abbreviations that are not international (DV%) and need to be explained. Is Bee pollen, and not bee bread, the one that revealed high phenolic and flavonoid contents. However, is BB that have synergetic effect when mixed with RJ in antioxidant and proteolytic power. BP only synergetic effect with RJ in antioxidant activity. Furthermore, there is nothing mentioned about the results in fortified fermented milk products.

Introduction:

In the second paragraph you say first that BP is the main source of protein for honeybee colony and 10 lines later it is written that BB constitutes the basic protein source for the bee colony. I know that both are “the same” product but is a bit confused to reader.

In the third paragraph is confused because is mixed about the proteolytic power of bee product and if the bee product can be hydrolyzed by various proteases and the functional activity of the bioactive peptides. I think are two different things. Also is the same problem in results (2.5).

Results and discussion:

2.1. Chemical composition: What normative do you use to obtain the DV? In EU are not the same values. Furthermore, I don’t know if it makes sence because nobody consumes 100g of BP, BB o RJ, Therefore, i.e. they could be a good source of Fe, Ca, Mg… but it not really provides that amount of minerals or proteins…

Potassium dominated BB mineral content is 3.02%

The last two sentences of the second paragraph: “Generally, comparing the nutritional….. reported by [7]” is not understandable.

2.2. Amino acids composition

In the references that I have the essential and conditionally essential amino acids are not the same that the authors described. Essential Aa are Leucine, isoleucine, lysine, methionine, phenylalanine, threonine, valine and tryptophan, while the conditionally essential are histidine, cysteine, tyrosine and arginine. Why Tryptophan was not analysed?

BP contents more aspartic acid than proline, why do you show higher importance to proline? How do you obtain their AAS (this values are not in the table).

In BB the limiting amino acid is threonine (that have the lowest AAS) or histidine (that have lower mg/g)?

The last sentences in this part “Consequently, mixing complementary protein…. Long-term adequacy” is a bit messy.

2.3. LC-MS-MS analysis

For me is a bit confused this part. What information is evaluated in the networking GNPS? Directly the spectra after LC-MS-MS? More comments about it in Table 3.

2.4. Antioxidant potentials

In the first paragraph exist repetition about the idea of BP highest total phenolic content….

In the second paragraph “which is still less than that with individuals” seems to say that have lower antioxidant activity and not lower IC50, which is the contrary.

2.5. Protease activity

Are those very small diameters significant? Can they be seen? In the Figure 3 I can’t see any halo.

The last sentence in the page 13 “Bee bread/royal jelly and tri….. proteases activities” and the First sentence in the page 14 about pH 5 is repeated.

The proteolytic activity of RJ are supported by references about hydrolyzed by enzymes of the larvae. I think that is not the same.

2.6. SDS-PAGE of honeybee

The sentence “ while the protein with 23.6 KDa corresponding to papain according to “ is not finished. In what honeybee product was found that protein?

2.7. Physicochemical characteristic

There is any mention to  the different sugar concentration that you explain in material and method.

2.8. LAB starter culture

The counts of microorganism normally are presented in log CFU/g.  The last sentence in the first paragraph doesn’t make sense there.

The first sentence in the second paragraph what mean? What could explain the elevation in counts?

Streptococcus thermophiles in cursive

The last part “The obtained pattern differed from what reported by….. 15 days of cold storage” I get lost in reading.

2.9. Sensory evaluation

I think that the Figrue 7 that is mentioned in mid paragraph TRJ is Figure 6

Materials and methods:

 I don’t know if more information about protein and total sugar method could be described.

In minerals analysis what do you used for dilute?

Gradient in LC-MS-MS analysis?

5-min isocratic run 5% CAN and 0.05% FA is repeated.

Reference Sakanaka, Tachibana.. (2005) is not with number. Is AlCl3 used?

DPPH is not the best radical to assay. Why do you select it?

In the proteolytic activity is described “formation of a clear zone around colonies”. What colonies? Maybe say that first qualitative analysis was performed by agar and later quantitative assay… Why do you use different pH?

What is DDT? SNF?

The preparation of honeybee product-fortified fermented dairy product for me is not good enough. In which concentration BB, BP and RJ were added?  1 and 7%? In the results there is not mention about it.  What mean sugar solution was added? The sugar solution is 1 and 7%?

References:

I have found two articles related with the manuscript that are not mentioned and could be appropriate:

-          Royal jelly improves the physicochemical properties and biological activities of fermented milk with enhanced probiotic viability, 2022, LWT

-          An evaluation of milk enriched with royal jelly. Milchissenschaft, 2002

Figures and Tables:

Some tittles of Figures are not good described. i.e. Figure 1B maybe is better flavonoids found in bee product…

Table 3: There are some metabolites described in 2 (BB) and in the figure is in 3 (BP) or vice versa, for example catechin, decanoyllysolecithic or why some of them have two numbers and in the figure only one color? Why are in bold font three metabolites? Why not all the metabolites are in the Figures 1A, 1B and 1C? For example Kaempferol? There are some parent mass (m/z) that are not the same that in the figure, i.e. jelleine II and Royalisin.

Author Response

Response letter enclosed

Reviewer 2 Report

Dear authors,

Some of my comments to your interesting research is attached. 

Kind regards,

Author Response

Response letter enclosed
